# Design of Current Equalization Circuit in Dual Ethernet Power Supply System

Xingyu Guan, Xinyuan Hu, Junkai Zhang and Yanfeng Jiang *

Department of Electrical Engineering, School of IoT Engineering, Institute of Advanced Technology,
Jiangnan University, Wuxi 214122, China; 6221916018@stu.jiangnan.edu.cn (X.G.);
6221916019@stu.jiangnan.edu.cn (X.H.); 6221916045@stu.jiangnan.edu.cn (J.Z.)
* Correspondence: jiangyf@jiangnan.edu.cn

**Abstract:** A current-balancing circuit for a dual-channel Ethernet power supply system is designed in this paper, which can be used to solve the mismatch between the two channels caused by unavoidable factors, such as mismatched resistances, temperatures and voltages. Based on the design, the mismatch of the currents between the two power transmission paths can be controlled to be less than 1% of the original ones. It can be operated under these conditions with the changes of the load current and the PSE output voltage. The maximum output power of the dual-channel power supply can reach up to 96.5 W. When the DC–DC conversion efficiency is less than 75%, it can still provide 72 W for the PD end, meeting the requirements of the PoE power system. The current-balancing circuit designed in the paper has potential application value to improve the dual PoE power supply system.

**Keywords:** Power over Ethernet (PoE); current balancing; MOSFET rectifier bridge; power supply sensitivity; temperature sensitivity

## 1. Introduction

Power over Ethernet (PoE) is an advanced technology that utilizes an Ethernet cable to realize the dual transmission of power and data. It is compatible with the existing Ethernet cabling infrastructure. The power system can be installed easily, which can reduce the cost required for equipment installation and maintenance [1].

The PoE system consists of power supply equipment (PSE) and power receiving equipment (PD) [2] to realize real-time monitoring of various power consumption in the power grid. PSE is a device composed of Ethernet switches and hubs, which transmits power and data signals to the twisted pair of the LAN through the hub. The PSE equipment undertakes the important responsibilities of PD query, power supply and power planning management. The PD control chips that meets IEEE802.3af/at standards have been promoted by Texas Instruments, Linear Technology and Maxim [3–8]. Compared to [3–7], a power MOSFET is integrated in the design proposed in Ref. [8]. However, because it only supports a low output power (6.49 W), the application range is limited. In order to meet various performance requirements of PD, Li Yongyuan et al. proposed an integrated PoE interface and DC–DC controller, which was optimized for the isolated converter to achieve a precise current limitation, a high power efficiency, a high voltage accuracy and a fast load step response [9].

In 2003, the IEEE 802.3af standard was formulated by the PoE Group of the Institute of Electrical and Electronics Engineers (IEEE) and includes two linear power supply modes [10]. The first mode is to supply power to the idle pair, in which 4|5 and 7|8 are two pairs of cable lines for power transmission, with the provision of 4|5 for the positive pole and 7|8 for the negative pole. The other mode is to supply power to the data pair. The 1|2 and 3|6 twisted pairs are used for power transmission. The power supply polarity of the mode is arbitrary [10]. With the wide application of wireless LAN in offices and

homes, Zargari, M et al. proposed a single-chip dual-band three-mode CMOS transceiver that supports IEEE 802.11a/b/g WLAN standard [11].

The IEEE 802.3af standard can provide the maximum available power of 12.95 W to the RJ45 socket at the input end of the receiving device. This limitation prevents PoE from powering large power application scenarios. Since industrial products require greater power and higher conversion efficiency [12], it is necessary to provide higher power levels. On the basis of a compatibility with IEEE 802.3af, the IEEE 802.3at standard was proposed in 2009, which increased the maximum usable power from 12.95 W to 25.5 W [13].

To meet the demand for higher power in applications such as Wi-Fi access points, surveillance cameras and connected LED lighting, the industry has published a new IEEE 802.3bt standard [14,15], as shown in Figure 1. At the same time, non-IEEE standard methods such as CISCO's [16] or LTPOE++ [17] of UPOE Linear Technology have been published [18].

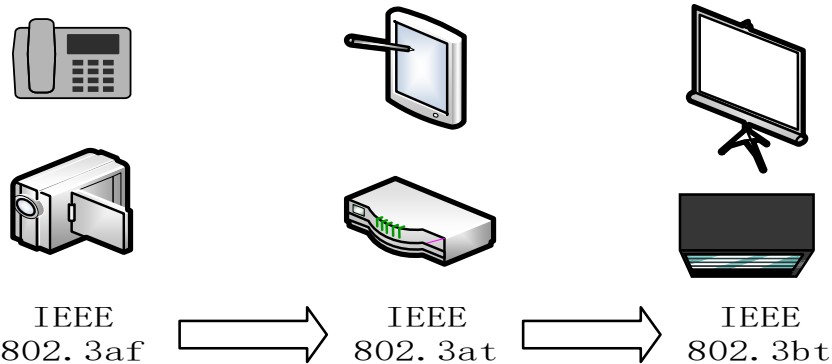

**Figure 1.** IEEE development process.

In 2012, Wu Jiande's team of Zhejiang University proposed a method to add loops in PoE infrastructure and use two DC–DC converters to achieve current balance [19]. Huang Jianyu et al. proposed a 90 W high-power Ethernet power supply system [20]. In 2017, Xiao Zhiming et al. proposed a balancing circuit based on MOS rectifier diodes [21]. In practice, a mismatch [22] between the cables, connectors and components of the two power conduction paths causes the maximum available power to be derated, or the current may flow backward from one pair of cables to the other. Therefore, how to balance the two currents [23,24] is the problem that needs to be solved for the dual-channel PoE system.

A new two-channel current-balancing circuit is designed in the paper, which has an obvious effect on improving the reliability of a two-channel PoE power supply system. The contents of the paper is organized as follows: the second part focuses on the dual-power supply Ethernet network modeling analysis, the third part mainly introduces the design of the dual equalizer circuit and simulation results, the fourth part introduces the overall simulation results and experimental results. The fifth part gives the main conclusions.

## 2. Dual-Power Supply over Ethernet

### 2.1. Steady-State Modeling and Analysis of PoE System

The schematic of the PSE-PD interface is shown in Figure 2a. The ports include four transformers T1–T4, Ethernet cables $P_{12}$–$P_{78}$, two rectifier bridges for adjusting the power polarity of the Ethernet cable, and two for the PD connection to the PoE system. According to the IEEE 802.3af/at standard, the information data are transmitted through the transformer T1–T4 of the data pair $P_{12}$ and $P_{36}$, and PSE can use the same data pair or spare $P_{45}$ and $P_{78}$ to provide power to the load through the diode bridge [25].

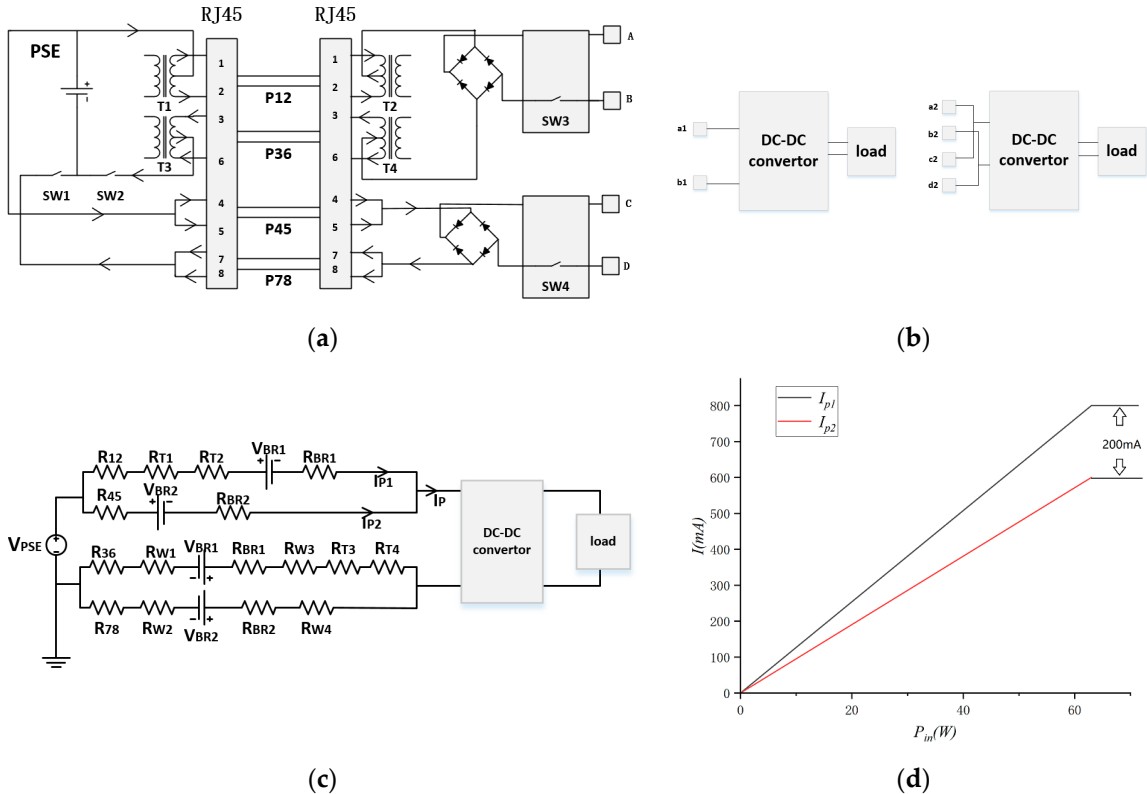

**Figure 2.** (**a**) Schematic of the PSE−PD interface structure; (**b**) schematic of the PD interface structure; (**c**) equivalent circuit of the transmission path structure; (**d**) relationship between the current and the power of the two channels.

In order to reduce the transmission energy consumption and packet delay [26] to increase the power capability, the method of dual power supply is adopted to connect ports A, B, C and D to ports a2, b2, c2 and d2 corresponding to the PD interface structure in Figure 2b.

For the dual power supply structure [19], its equivalent circuit, including the parasitic resistances, is shown in Figure 2c. In the figure, $R_{12}$ and $R_{36}$ denote the cable resistances from data to $P_{12}$ and $P_{36}$, respectively. $R_{45}$ and $R_{78}$ denote the cable resistances of spare data to $P_{45}$ and $P_{78}$, respectively. $R_{T1}$–$R_{T4}$ are the transformer resistances. $R_{BR1}$ and $R_{BR2}$ are diode bridge resistors. $R_{w1}$ and $R_{w2}$ are resistors on switches $S_{w1}$ and $S_{w2}$ in the PSE. $R_{w3}$ and $R_{w4}$ are on-resistors on switches $S_{w3}$ and $S_{w4}$. $V_{BR1}$ and $V_{BR2}$ are the forward voltage drops on the two diode bridges. $V_{PSE}$ is the voltage on the PSE.

If the sum of the two currents is IP, and the currents on path 1 and path 2 are IP1 and IP2, respectively, we obtain:

$$\begin{cases} I_{P1} \times (R_{12} + R_{T1} + R_{T2} + R_{BR1}) + V_{BR1} \\ = I_{P2} \times (R_{45} + R_{BR2}) + V_{BR2} \\ I_{p1} + I_{p2} = I_P. \end{cases} \tag{1}$$

In order to simplify the analysis, the resistance of the two paths is represented by RA and RB, respectively, and the specific expression is as follows:

$$\begin{cases} R_A = R_{12} + R_{BR1} + R_{T1} + R_{T2} \\ R_B = R_{45} + R_{BR2} \end{cases} \tag{2}$$

From Equations (1) and (2), the expressions of IP1 and IP2 can be obtained as follows:

$$\begin{cases} I_{P1} = \dfrac{V_{BR2} - V_{BR1}}{R_A + R_B} + \dfrac{R_B}{R_A + R_B} I_R \\ I_{P2} = \dfrac{V_{BR1} - V_{BR2}}{R_A + R_B} + \dfrac{R_A}{R_A + R_B} I_R \end{cases} \tag{3}$$

According to the IEEE standard, the maximum value of the cable resistance mismatch is 3% [27]. In the circuit shown in Figure 2c, $V_{BR1} = 0.65$ V, $V_{BR2} = 1$ V, $R_A = 9.84$ $\Omega$, $R_B = 7.71$ $\Omega$ and the maximum current difference under different load conditions can be calculated. The relationship between the loop current and the input power is shown in Figure 2d. The results show that the average current per path is about 700 mA, and in the worst case, the maximum current difference can be up to 200 mA. The imbalance current may cause one of the paths to have an overcurrent, which would be shut down by the interface circuit. Eventually, the other path may also exceed the current limit. The PD may be shut down under the influence of the imbalance current.

### 2.2. PD Interface Structure

Based on the traditional PD interface structure, MOSFET transistors were used to replace the rectifier diode in the rectifier bridge in the designed interface system proposed in this paper [28]. Its schematic is shown in Figure 3a, including the RJ45 interface, bridge rectifier current, current balancing module, bias and bridge control module and signature and classification module. The signature and classification modules are built according to IEEE 802.3at standard [13]. The power transmission path of $P_{12}$ and $P_{36}$ (data pair) is called channel 1, while the power transmission path through $P_{45}$ and $P_{78}$ (idle pair) is called channel 2.

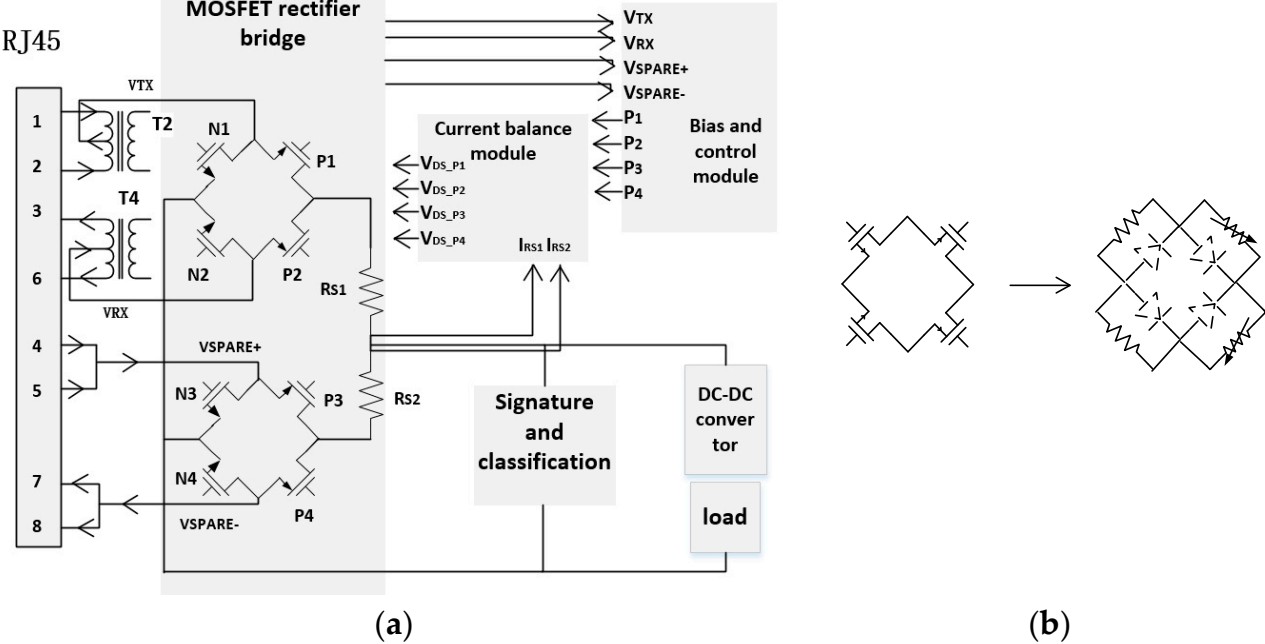

**Figure 3.** (**a**) Schematic of the PSE−PD interface structure. (**b**) Equivalentprinciple diagram of the rectifier bridge.

The schematic diagram of the MOSFET acting as a rectifier device is shown in Figure 3b. In order to ensure the reliable initial starting of the circuit, the parasitic body diode of the MOSFET is placed in the same direction as the conventional diode rectifier bridge [29]. The gate voltage of PMOSFETs is regulated, and their channel resistance can be regulated by a current balancing module. Thus, the active PMOSFETs in the diode bridge module are modeled by the parallel adjustable resistors and body diodes in Figure 3b. The on-

resistance of the MOSFET can be adjusted by changing the gate voltage in the current balancing module.

In the traditional rectifier bridge circuit, each diode has a certain voltage drop, which could increase the power loss. However, the MOSFET rectifier bridge shows a lower on-resistance value compared with the rectifier diode, which can effectively reduce the power loss of the circuit [28].

### 2.3. Circuit Stability Analysis

In circuit design, power supply, process and temperature (PVT) are important factors that need to be paid attention to. In circuit design, the variations of the factors should be considered and be dealt with [30].

The power supply voltage sensitivity represents the degree to which the current source is affected by the change in the power supply voltage, denoted by $S(I_0, V_P)$ [31]. $I_0$ is the output current of the current source, and $V_P$ is the power supply voltage. The parameter $S(I_0, V_P)$ represents the relative rate of change and can be written as:

$$S(I_0, V_p) = \frac{\partial I_0 / I_0}{\partial V_p / V_p} = \frac{V_p}{I_0} \frac{\partial I_0}{\partial V_p} \tag{4}$$

where $\partial I_0 / I_0$ is the relative change in output current, $\partial V_p / V_p$ is the relative change in supply voltage, and $\partial I_0 / \partial V_p$ is the partial derivative of $I_0$ with respect to $V_P$.

When $V_{DD} >> V_{GS}$, the output current of the current mirror $I_0$ is proportional to the supply voltage $V_{DD}$. The power supply sensitivity can be changed to:

$$S(I_0, V_{DD}) = \frac{V_{DD}}{I_0} \frac{\partial I_0}{\partial V_{DD}} \tag{5}$$

In the traditional current mirror, the relative change in the output current is almost equal to the relative change in the supply voltage. The traditional current mirror cannot be used in the designed circuit because of its relative low sensitivity. Therefore, a reference current source with $S(I_0, V_{DD}) \approx 0.06$ used as threshold voltage was adopted in this paper. As shown in Figure 4, this current source was used to generate a reference current for the whole circuit; the specific principle is introduced in detail in Section 3.

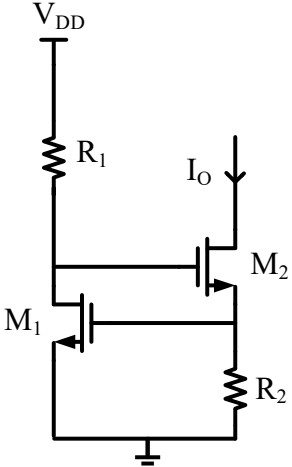

**Figure 4.** Reference current source using a threshold voltage.

In addition to the voltage sensitivity of the power supply, another important indicator of the current source is the temperature sensitivity. The smaller the temperature sensitivity, the less the circuit is affected by temperature in practical applications [22].

The temperature sensitivity of a current source is expressed as the relative change in the output current caused by each degree change in temperature. This change is called the relative temperature coefficient, which can be expressed as:

$$TC = \frac{1}{I_0}\frac{\partial I_0}{\partial T} \tag{6}$$

## 3. Design of Current-Balancing Circuit in Dual Ethernet Power Supply System

The control circuit of the rectifier bridge is used to control the switching of the rectifier MOSFETs in the rectifier bridge according to the power polarity of the output of the PSE end to ensure the normal operation of the PD end load. With the sampling of the two currents, the sampling resistance transmits the voltage difference to the current balancing module. The current balancing module and the finishing bridge control module can work together to control the gate voltage of the PMOS in the rectifier bridge with a high current and then change its on-resistance to balance the currents of the two channels.

### 3.1. Design of Bias Circuit Module

As shown in Figure 5a, a voltage $V_Z$ of 5 V and a stable reference current $I_O$ of 1.5 μA can be generated by the bias module for the entire current equalization circuit. According to the analysis in Section 2.3, the circuit uses a current mirror to replicate I1, and the copied current is only related to the width-to-length ratio of the MOS and is not sensitive to the interference of external factors such as temperature, which is conducive to the stable operation of the circuit in practical applications. The breakdown voltage of the regulator diode $D_Z$ is 5.6 V. Thus, the gate voltage of $M_{N5}$ is clamped at 5.6 V. A voltage $V_Z$ of 5 V can be produced based on $M_{N5}$ and Dz. $M_{N6}$, $M_{N7}$, $R_1$ and $R_2$ constitute the reference current source using the threshold voltage, and the current generated by it can be expressed as:

$$I_O = \frac{V_{GS6}}{R_1} = \frac{V_t + V_{ov6}}{R_1} \tag{7}$$

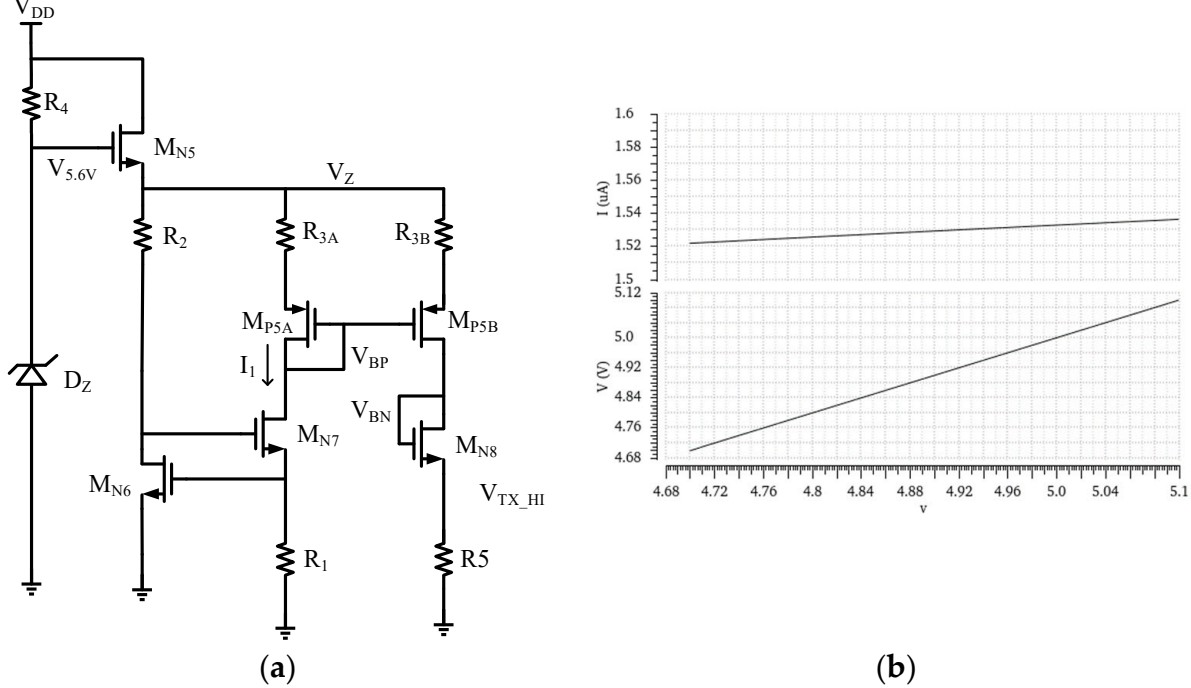

(a) (b)

**Figure 5.** (**a**) Schematic diagram of the bias module. (**b**) The reference current source output using the threshold voltage varies with the voltage.

When the $M_{N6}$ overdrive voltage $V_{OV6}$ is small relative to the threshold voltage $V_t$, the output current is mainly determined by the threshold voltage $V_t$ and $R_1$. In order to obtain a more stable output current, the aspect ratio of $M_{N6}$ is set to be 16:1, $R_1 = 500$ kΩ, and the output current $I_O \approx 1.5$ μA.

The bias $V_Z$ in Equation (7) can be calculated. According to the MOSFET overdrive voltage Equation (8) (*Kn6* is the conductivity coefficient), Equation (9) can be obtained:

$$V_{ov6} = \sqrt{\frac{I_{IN}}{K_{n6}}} \tag{8}$$

$$\frac{\partial I_O}{\partial V_Z} = \frac{1}{R_1}\frac{\partial V_{OV6}}{\partial V_Z} = \frac{1}{R_1}\frac{\partial}{\partial V_Z}\sqrt{\frac{I_{D6}}{K_{n6}}} = \frac{V_{OV6}}{2I_{D6}R_1}\frac{\partial I_{D6}}{\partial V_Z} \tag{9}$$

$I_{D6}$ is the source leakage current of $M_{N6}$.

Combining Equation (9) with Equation (5), the power sensitivity of the current source with the threshold voltage can be expressed as:

$$S(I_O, V_Z) = \frac{V_Z}{I_O}\frac{V_{OV6}}{2I_{D6}R_1}\frac{\partial I_{D6}}{\partial V_Z} = \frac{V_{OV6}}{2I_OR_1}\left(\frac{V_Z}{I_{D6}}\frac{\partial I_{D6}}{\partial V_Z}\right) = \frac{V_{OV6}}{2V_{GS6}}S(I_{D6}, V_Z) \tag{10}$$

The threshold voltage $V_T$ of $M_{N6}$ is 794 mV, and the overdrive voltage $V_{OV6}$ is 70 mV. $S(I_{D6}, V_Z) \approx 1.5$.

The power sensitivity is:

$$S(I_O, V_Z) = \frac{70}{2 \times (794 + 70)} \times 1.5 = 0.06 \tag{11}$$

The stability of the circuit was verified by a simulation. Figure 5b shows the simulation results. With $V_Z$ changing from 4.7 V to 5.1 V, the variation in the output current is within 0.01 μA. The key parameters of the bias circuit (Figure 5a) are listed in Table 1.

**Table 1.** Key parameters of the bias circuit.

| Device | $R_1$ | $R_2$ | $R_{3A-B}$ | $R_4$ | $R_5$ |
|---|---|---|---|---|---|
| **Dimension (KΩ)** | 500 | 50 | 500 | 1000 | 500 |
| **Device** | $MN_5$ | $MN_6$ | $MN_7$ | $MN_8$ | $MP_{5A-B}$ |
| **Dimension (W/L)** | 20 | 16 | 8 | 4 | 4 |

*3.2. Design of Rectifier Bridge and Control Circuit Module*

As shown in Figure 6a, a control circuit of a MOSFET rectifier bridge was designed. $V_{TX}$ and $V_{SPARE+}$ are the voltages used by the Ethernet cable to transmit power to the two power transmission paths. The control signals $V_{TX\_HI}$, $V_{RX\_HI}$, $V_{SP\_HI}$ and $V_{SP+\_HI}$ are used to control $M_{P6A-D}$ and $M_{N11A-D}$. The output currents $I_{N1-4}$ and $I_{P1-4}$ can be used to change the gate voltages of the MOSFETs in the bridge. The PMOS on the high-voltage side and the NMOS on the low-voltage side can be switched on to achieve the effect of rectification.

The voltage regulator diode $D_Z$ stabilizes the gate voltages of $M_{N10A}$ and $M_{N10B}$ at 5.6 V. When $V_{TX}$ and $V_{SPARE+}$ have a high voltage, the two NMOS are turned on and the voltage of the input inverter is about 5 V. The threshold voltage of the inverter should be about 2.5 V.

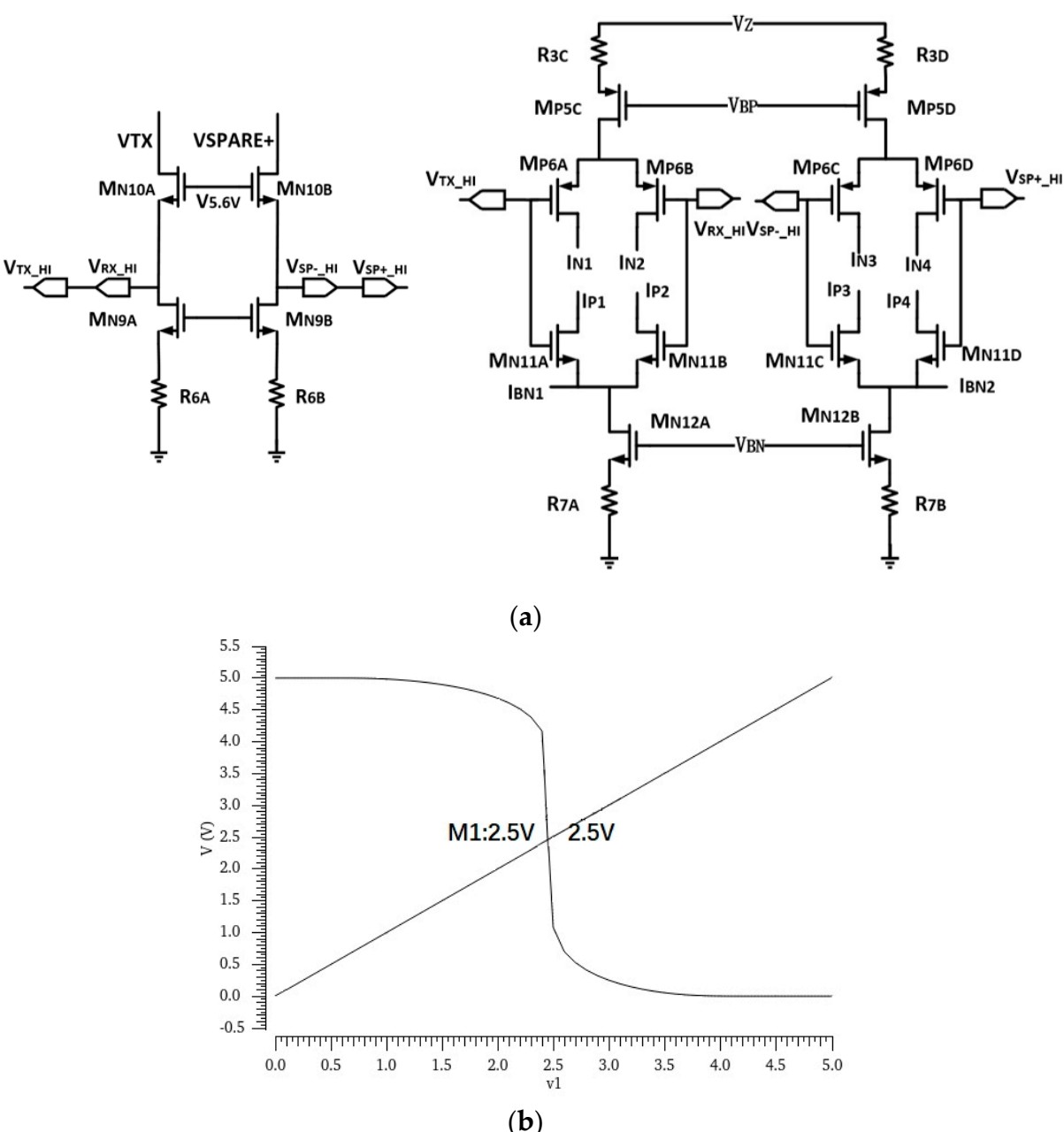

**Figure 6.** (**a**) Rectifier bridge control circuit. (**b**) Characteristic curve of the inverter.

When $(W/L)_p = 4(W/L)_N$, $r \approx 1$, the switching threshold of the inverter is set to be 2.5 V. The simulation waveform of the inverter is shown in Figure 6b.

The MOSFET rectifier bridge circuit [32] is shown in Figure 7a. $V_{TX}$, $V_{RX}$, $V_{SPARE+}$ and $V_{SPARE-}$ represent the power transmitted by the cable. $I_{P1-4}$ and $I_{N1-4}$ are the control currents of the finishing bridge control module. The on-gate source voltage of the MOSFET in the rectifier bridge can be expressed as:

$$V_{GS\_NPASS} = 2V_{GS\_MN6}/R_1 \cdot R_{N1-4} \tag{12}$$

$$V_{GS\_PPASS} = 2V_{GS\_MN6}/R_1 \cdot R_{P1-4} - I_{BN1,2} \cdot R_{P1-4} \tag{13}$$

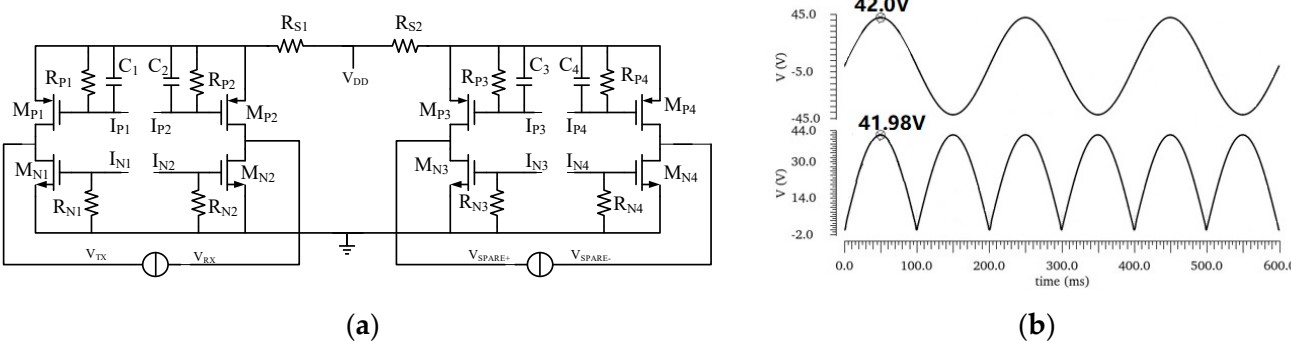

**Figure 7.** (**a**) MOSFET rectifier bridge circuit. (**b**) Rectifier bridge simulation results.

It can be seen that the $V_{GS}$ of the conducting MOSFET in the rectifier bridge is changed with the $V_{GS}$ of $M_{N6}$. It is not sensitive to the change in temperature anymore.

$I_{BN1,2}$ is the regulating current output of the current balancing module, which is used to balance the current in the two transmission paths. In order to reduce the power loss of the rectifier bridge, the on-conducting MOSFETs should be operated in the deep triode region to reduce their on-resistance (except for PMOS used to regulate the current imbalance). $M_{P5A-D}$ consists of a current mirror to replicate the current. The current value of each path in the current mirror is determined by the resistance value of the resistor. $R_{3A,B}$ is 500 Ω and $R_{3A,B}$ is 250 Ω. The source leakage current of $M_{P5C,D}$ is 3 µA. The current through the 4.8 MΩ resistance can produce a voltage drop of about 15 V, which shows enough margin to balance the current of the two paths.

When $V_{TX}$ is high, $M_{P1}$ and $M_{N2}$ are on-conduction, while $M_{P2}$ and $M_{N1}$ are cut off. When $V_{RX}$ is high, $M_{P2}$ and $M_{N1}$ are on, while $M_{P1}$ and $M_{N2}$ are off. As shown in Figure 7b, the input is AC and the output is DC, which ensures that the load side can be operated normally. Based on the simulation results, the maximum input voltage is 42 V. The maximum output voltage is 41.98 V. With the 0.02 V difference, the power loss can be obviously reduced compared with the traditional diode rectifier bridge.

The key parameters in Figures 6a and 7a are listed in Table 2.

**Table 2.** Key parameters in the rectifier bridge and bridge control circuit.

| Device | $R_{S1-2}$ | $R_{3C-D}$ | $R_{6A-B}$ | | |
|---|---|---|---|---|---|
| Dimension (Ω) | 0.8 | 250 | 500 | | |
| Device | $M_{P5C-D}$ | $M_{P6A-D}$ | $M_{N9A-B}$ | $M_{N10A-B}$ | $M_{N11A-D}$ |
| Dimension (W/L) | 4 | 20 | 4 | 20 | 20 |

### 3.3. Design of Current Balancing Module

Figure 8a shows the working principle of the current-balancing circuit in a dual power-over-Ethernet system. Assuming that the inputs of $V_{TX}$ and $V_{PSARE+}$ are high, and $V_{TX} > V_{SPARE+}$, one of the four PMOSs in the bridge rectifier circuit is used to adjust the two currents. The remaining on-conducting MOSFETs operate in the deep transistor region to reduce the power loss of the bridge. According to the input voltage polarity, $M_{P1}$, $M_{P3}$, $M_{N2}$ and $M_{N4}$ in the rectifier bridge are turned on, while $M_{P2}$, $M_{P4}$, $M_{N1}$ and $M_{N2}$ remain off. All MOSFETs except MP1 operate in the deep transistor region, and they can be equivalent to resistors.

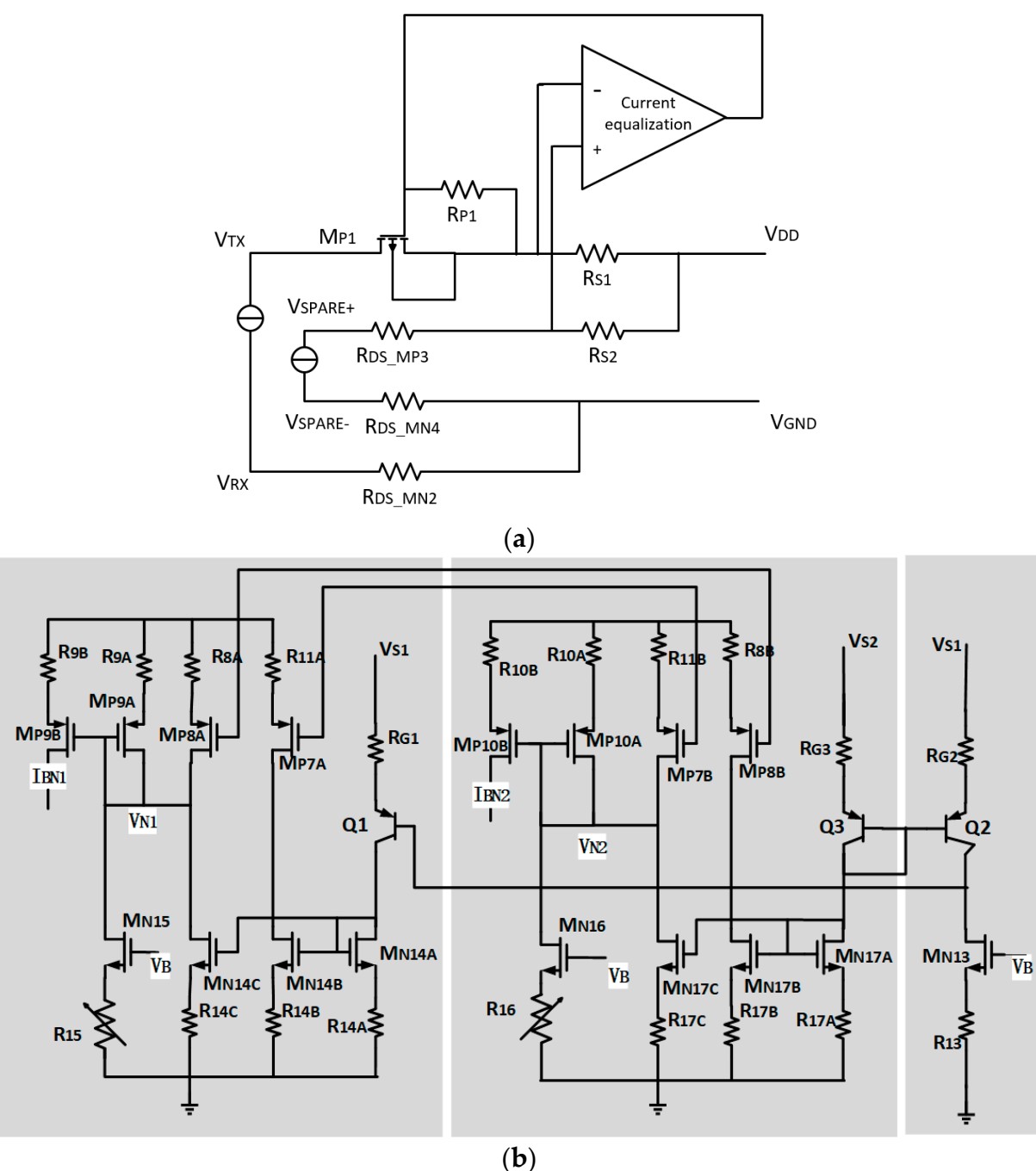

**Figure 8.** (**a**) Schematic diagram of the current-balancing circuit. (**b**) Designed current−balancing circuit.

In the case of an unexpected condition, assuming the current in channel 1 is greater than that in channel 2, the channel 1 is closed by the current limiting protection device in the Ethernet power supply system due to excessive current, resulting in a significant reduction in output power. In order to equalize the two currents, the resistors $R_{S1}$ and $R_{S2}$ sample the currents of the two paths and convert the current difference into a voltage difference. The voltage difference is transmitted into the current equalization module. The current equalization module outputs the corresponding regulation current $I_{BN1}$ through the resistance $R_{P1}$. The gate voltage of $M_{P1}$ is reduced to convert it from the deep triode region to the triode region or the saturation region. The on-resistance is increased, and the $V_{DS}$ of $M_{P1}$ is changed to balance the two currents.

According to the IEEE 802.3at standard, the maximum length of the cable is 100 m, the maximum resistance tolerance is $\pm 3\%$ [23], and the maximum current value of each transmission path is 1 A, so the maximum voltage difference between the PSE cable and the PSE-PD's two interfaces is about 250 mV. Due to the actual application of the two circuits used in the device, a comprehensive consideration leads to a deviation of about 0.4 V in the two paths. The on-voltage of the $M_{P1-4}$ body diode used in the bridge rectifier circuit in this design is about 0.7 V, which can change the current mismatch caused by the maximum 0.7 V voltage in the two paths and has enough ability to change the 0.4 V deviation in a practical application.

The schematic diagram of the current-balancing circuit is shown in Figure 8b. $R_{S1}$ and $R_{S2}$ perceive the current difference in the two channels and convert the current into voltage. The currents are transmitted to the $V_{S1}$ and $V_{S2}$ ports of the current balancing circuit, respectively. $Q_1$ and $Q_3$ are common base amplifier circuits, which have current-following and good high-frequency characteristics. A bipolar transistor shows a smaller impedance compared with a MOSFET, which can be better for the impedance coupling. $Q_2$ provides the same bias points for $Q_1$ and $Q_3$, while $Q_1$ is isolated from $Q_3$ to avoid any signal coupling between the two channels.

$\Delta I$ represents the current difference in the two transmission paths. The current difference between $Q_1$ and $Q_3$ is

$$\Delta I_{Q1,Q3} = \Delta I \cdot R_{S1,2} / R_{G1,3} \tag{14}$$

The current difference is added on the $V_{N1}$ and $V_{N2}$ routes by the replication of the current mirror. The current flowing through $M_{P9A}/M_{P10A}$ is increased/decreased depending on the polarity of the current difference. Current mirrors $M_{N14A-C}$ and $M_{N17A-C}$ can duplicate the change in the current of $Q_1$ and $Q_3$, while $M_{P7A-B}$ and $M_{P8A-B}$ duplicate the change in the current of $Q_1$ and $Q_3$ on the other side, respectively.

When the current in channel 1 is greater than the current in channel 2, the voltage $V_{S1}$ is greater than $V_{S2}$. When the two currents are balanced, the collector currents of Q1 and Q3 increase and decrease, respectively. This change trend is reflected into $M_{N14B,C}$ and $M_{N17B,C}$, respectively, through the current mirror, so that the current flowing through $M_{N14B,C}$ increases. The current flowing through $M_{N17B,C}$ decreases, while the current variation trend in $M_{N14B}$ and $M_{N17B}$ decreases the current of $M_{P8A}$ and increases the current of $M_{P7A}$ after passing through the current mirror, respectively. Thus, the current flowing through $M_{P9A}$ and $M_{P10A}$ increases and decreases, and the current mirrors $M_{P9A,B}$, and $M_{P10A,B}$ replicate the increased current to $I_{BN1}$ and $I_{BN2}$ in a ratio of 1:5. $I_{BN1}$ and $I_{BN2}$ are inputted into the bridge control circuit in Figure 6a to increase and decrease $I_{P1}$ and $I_{P3}$, respectively. The decrease in current flowing through $R_{P1}$ causes the gate voltage of $M_{P1}$ to increase, the conduction resistance to increase and the $V_{DS}$ to increase to change the imbalance of the two currents. The increased $I_{P3}$ increases the $V_{GS}$ of $M_{P3}$ and continues to work in the deep transistor region, reducing the power loss of the circuit.

In practical applications, with the unavoidable errors of the devices, an error-regulating circuit is designed, as shown in Figure 8b. Two pull-down current paths $M_{N15}$ and $M_{N16}$ are placed on $V_{N1}$ and $V_{N2}$, respectively, to offset the errors caused by $R_{S1}$, $R_{S2}$, $R_{G13}$, $Q_1$, $Q_2$ and $Q_3$. The two pull-down trimmers are adjusted by adjusting the resistance values of the adjustable resistors $R_{15}$ and $R_{16}$. $M_{P9A}$ and $M_{P10A}$ are connected in the form of diodes, so the impedance on $V_{N1}$ and $V_{N2}$ is relatively small. Because the impedance of the $M_{N15}$ and $M_{N16}$ pull-down current path is high, the frequency response of the original current balance circuit is not affected by the error-regulation circuit.

In the rectifier bridge circuit as shown in Figure 8b, the capacitor $C_{1-4}$ is used to compensate the stability of the loop. During each current-equalizing process, only one

MOSFET in $MP_{1-4}$ is used to adjust the loop current. The current difference between the two paths after current balancing is:

$$\Delta I_{\text{closed}} = \frac{(\Delta V_{\text{PSE}} + 0.5\Delta R_{\text{CH}} \cdot I_{\text{LOAD}} + \Delta V_{\text{DS}})/R_{\text{CH1.2}}}{1 + T_{\text{L}}} \tag{15}$$

where $\Delta V_{PSE}$ is the voltage difference between the PSE end and the two paths, $\Delta R_{\text{CH}}$ is the cable resistance difference, and $\Delta V_{\text{DS}}$ is the drain-to-source voltage difference of the MOS device in the rectifier bridge. After the adjustment of the current-balancing circuit, the current mismatch between the two channels is reduced.

Figure 9 is a simulation in the worst-case scenario, with RCH1 and RCH2 being 0.5 $\Omega$ and the voltage difference between the two channels being 0.4 V. The low-frequency gain of the circuit is 52 dB, the gain is 400, and the phase margin is about 90°.

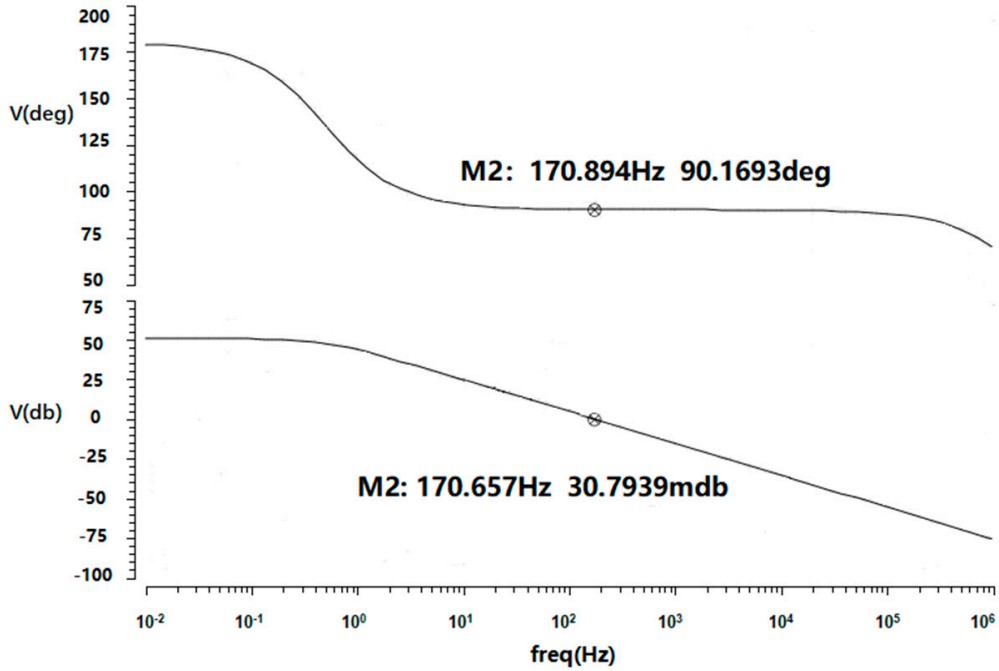

**Figure 9.** Simulation results of current−balancing circuit.

Key parameters in Figure 8b are listed in Table 3.

**Table 3.** Key parameters of the devices in the current-balancing module.

| Device | $R_{G1-3}$ | $R_{8A-B}$ | $R_{9A}$, $R_{10A}$ | $R_{9B}$, $R_{10B}$ | $R_{11A-B}$ | $R_{13}$ | $R_{14A-C}$ | $R_{15,16}$ | $R_{17A-C}$ |
|---|---|---|---|---|---|---|---|---|---|
| **Dimension (KΩ)** | 1 | 50 | 50 | 10 | 50 | 40 | 50 | 50 | 50 |
| **Device** | $M_{P7A,B}$ | $M_{P8A,B}$ | $M_{P9A,B}$ | $M_{P10A,B}$ | $M_{N13}$ | $M_{N14A-C}$ | $M_{N15}$ | $M_{N16}$ | $M_{N17A-C}$ |
| **Dimension (W/L)** | 4 | 4 | 4 | 4 | 12 | 4 | 4 | 4 | 4 |

## 4. Results and Discussion

### 4.1. Dual Ethernet Power Supply Current-Balancing Circuit

Figure 10 shows the two currents under the condition that the voltage difference increases from 0 V to 40 mV without the current-balancing module. When the voltage difference is 400 mV, the current difference between the two channels is 321 mA. In practical applications, the large current will be turned off by the current limiting circuit in the Ethernet power supply system, affecting the normal power supply of the entire system.

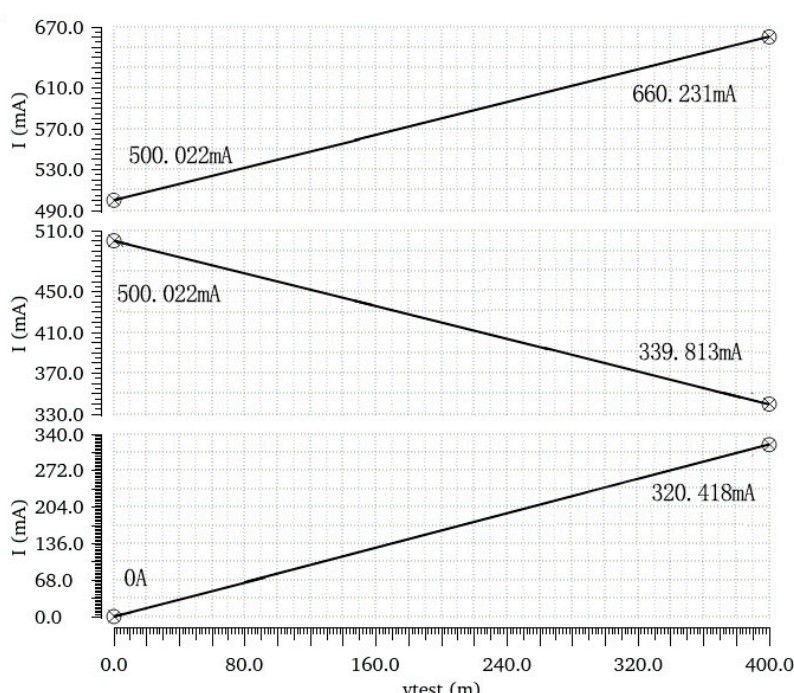

**Figure 10.** Influence of different voltage differences on the two currents.

Figure 11a,b show the simulation results of the non-current-balancing circuit and the current-balancing circuit. Adding a voltage difference of 200 mV and 400 mV at 0.4 s and 0.7 s (assuming that the voltage in channel 1 is greater than that in channel 2) results in a current difference of 161 mA and 321 mA, respectively, in the absence of a current balancing module. With the introduction of the current-balancing module, the current difference is reduced to 0.65 mA and 0.91 mA at 200 mV and 400 mV, and the current mismatch is reduced to 0.2% without any equalization of voltage differences up to 400 mV. Figure 11c shows the situation when the voltage in channel 2 is higher than that in channel 1. The current equalization circuit of the designed dual power supply Ethernet system can be used when the PSE input is of different power polarity.

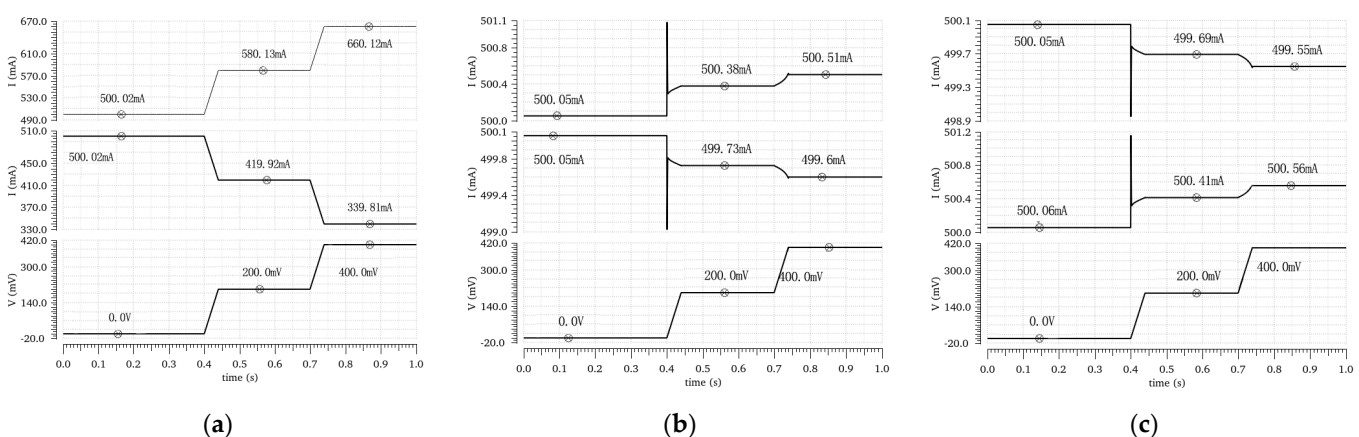

**Figure 11.** (**a**) Currents without current−-balancing module. (**b**) Currents with current−balancing module. (**c**) Currents on condition of larger voltage of channel 2 than channel 1.4.2. Disturbance Stability of Two-Channel Ethernet Power Supply Current Equalization Circuit

In order to further test the reliability of the current equalization circuit of the dual Ethernet power supply, the conditions of input common-mode voltage variations and load current variations were simulated.

Under the conditions of the voltage difference of the input common-mode voltage of 400 mV, Figure 12a,b show the currents of the two channels with or without the current-balancing circuit. When there is no current-balancing circuit, the current difference between the two channels is 375.66 mA, as shown in Figure 12b. On the contrary, with the current-balancing circuit, the current difference between the two channels can be reduced to 0.75 mA, as shown in Figure 12a. As shown in Figure 12c, when the load current is 600 mA, the current of channel 1 is 300.44 mA, and the current of channel 2 is 299.67 mA, the current difference between the two paths is 0.77 mA. At 550 ms, the load current is 900 mA. At that time, the current of channel 1 is 450.49 mA, and the current of channel 2 is 449.62 mA. The current difference between the two channels is 0.87 mA.

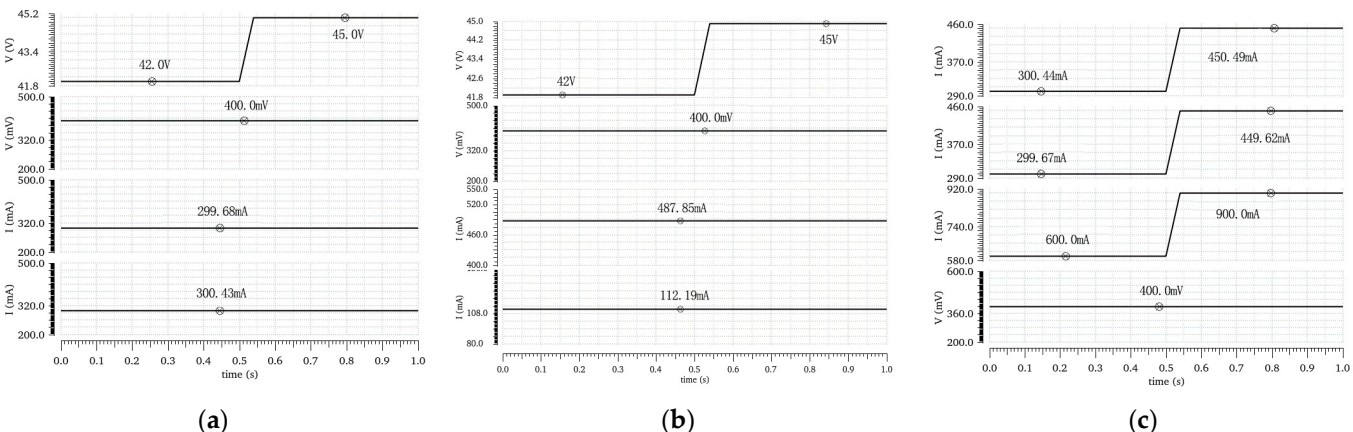

(**a**)　　　　　　　　　　　(**b**)　　　　　　　　　　　(**c**)

**Figure 12.** (**a**) Currents of the two channels with the current-balancing circuit. (**b**) Currents of the two channels without the current-balancing circuit. (**c**) Currents of the two channels with different loads.

*4.2. Analysis of Output Power and Power Loss of Dual Ethernet Power Supply Current-Balancing Circuit*

In order to verify the maximum output power and power loss of the circuit, the output current was varied from 0.6 A-2 A. Based on the IEEE standard, the maximum current per transmission channel is 1 A. Figure 13 shows that the output current is changed from 0.6 A to 2 A, and the output power is changed from 29.3 W to 96.5 W, accordingly. Assuming a DC–DC conversion efficiency of 75%, the maximum output power can provide 72 W of power to the PD end. The power loss is increased from 500 mW to 2.86 W.

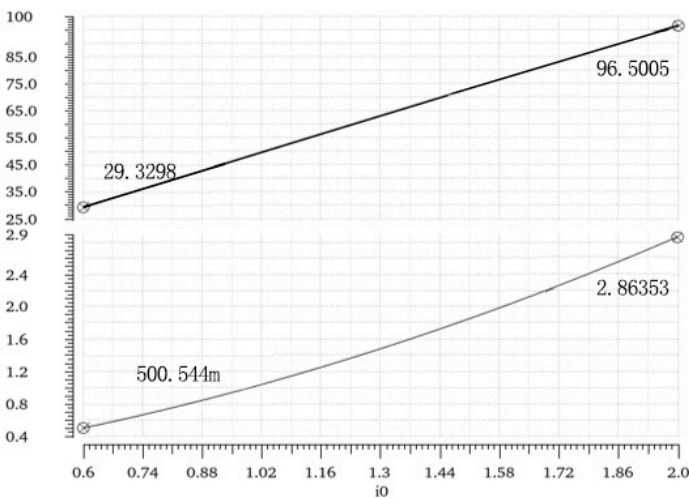

**Figure 13.** Analysis of the output power and the power loss.

## 5. Conclusions

PoE technology is popular in the market because of its advantages such as simplified wiring, energy saving and environmental protection. However, the traditional single-channel Ethernet power supply system cannot meet the high power demand of an electric appliance. The dual-channel Ethernet power supply system can provide more power. However, the imbalance of the two channels is one of the key problems and should be solved. In this paper, a current-balancing circuit in a dual Ethernet power supply system was proposed and designed. Comparison between the traditional and the designed systems is listed in Table 4. The simulation results showed that the circuit could solve a mismatch of up to 0.7 V between the two channels and reduce the current error in the two power conduction channels to be less than 1% of the error in the non-current-balancing circuit. It shows that the design can work stably under the condition of a load current change and PSE input voltage change. The maximum output power can reach up to 96.5 W. The power can provide about 72 W to the PD end when the DC–DC conversion efficiency is 75%.

**Table 4.** Comparison with and without current-balancing circuits.

|  | No Current-Balancing Circuit (400 mV Voltage Difference) | Current-Balancing Circuits (400 mV Voltage Difference) |
|---|---|---|
| Current error | 1 | 0.2% |
| Stability | 375.6 mA | 0.75 mA |
| Power | 25.5 W | 72 W |

**Author Contributions:** Methodology, X.G.; formal analysis, X.G.; writing—original draft, X.G.; formal analysis, X.H.; formal analysis, J.Z.; writing—review and editing, Y.J.; supervision, Y.J. All authors have read and agreed to the published version of the manuscript.

**Funding:** This research received no external funding.

**Data Availability Statement:** The data that support the findings of this study are available from the corresponding author upon reasonable request.

**Conflicts of Interest:** The authors declare no conflict of interest.

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
