# Peer review of "Design of Current Equalization Circuit in Dual Ethernet Power Supply System"

_jlpea, doi:10.3390/jlpea13040060_

Round 1

Reviewer 1 Report

Comments and Suggestions for Authors

This paper present the novel methodology for designing current equalization circuit in dual Ethernet power supply system. Even though the authors have put their sincere efforts to prepare this manuscript, I believe the manuscript could be more improved after revision. For this reason, I would suggest 'major revision'.

1) More than anything else, I feel that there is actually no comparison with the previous research. Therefore, if possible, please kindly add one more table which describes the comparison between your result and other group's previous result.

2) Fig. 1 is somewhat small. Please kindly make it larger so that the potential reader could understand with more ease.

3) In page 3, the following sentence needs some references. "According to the IEEE standard, the maximum value of the cable resistance mis- 94 match is 3%."

4) In page 3, there is typo. Please kindly revise it. "it is built following the 108 IEEE 802.3at standard [10]." -> IEEE 802.3 at standard [10].

5) In Table 2, the unit 'mOhm' should be located at the first row. Therefore, please kindly revise the word 'dimen-sion' as 'dimen-sion [m~~~]'  (~~~~ means roman character 'ohm'. Sorry I can't type roman character here.) 

Thanks.

Comments on the Quality of English Language

Please kindly find the above comments. Thanks.

Reviewer 2 Report

Comments and Suggestions for Authors

Dear Authors, there are some parts to improve/correct:

1) figure 2b, figure 2c, figure 2d: improve the quality of the figures. For the parts written both in the colored box and outside, use black characters for better readability.

2) Line 95-96: You wrote that in figure 2(c) RA= 9.84Ω, RB= 957.71Ω. I don't see any RA and RB. To be checked and corrected

3) Figure 2d: In the paper, describe well how the figure and this statement should be read "The results show that the average current per path is about 700 mA, and in the worst case, the maximum current difference can be up to 200 mA". To be verified and integrated.

4) Figure 3b: On the right, in the equivalent diagram of the "rectifier bridge circuit", do only two MOSFETs behave as variable resistors? To verify.

5) Eqn.5-6: What does Kn6 indicate? To be included in the description.

6) Table 1, Table 2, Table 3: The tables must be rewritten by putting (first line: device), (second line: dimension). If you do it this way, you don't need to repeat device and dimension in each column every time. In each column, indicate the device identifier. Example: first line with second column: R1, second line with second column: value, etc.

Best Regards

Comments on the Quality of English Language

The english is good

Reviewer 3 Report

Comments and Suggestions for Authors

1) The symbols in Figures 2 and 3 in this paper are very unclear. Please improve them.

2) The symbols used for I0 in equation (1) are inconsistent with Io in line 131, and they seem to be reused with IO in line 155, which is very easy to confuse. It is recommended that the author correct them together.

3) Please label the position of Io in Figure 4.

4) This paper only provides simulation results, and it is recommended that the author provide actual measurement results. Secondly, if component errors and temperature rise caused by long-term testing are added, the current error should not be lower than 1% as simulated results.

Reviewer 4 Report

Comments and Suggestions for Authors

1- figure 2 and 3 and 8 qualities are not good, please modify.

2- figure 6(b) was taken from another paper which is not acceptable, please plot the figure.

3- The balancing of the schematic diagram of figure 7 should be explained in more details.

4- the most recent literature should be considered in introduction and reference sections.

4-

Round 2

Reviewer 1 Report

Comments and Suggestions for Authors

The authors have put their sincerely efforts into revising the manuscript. Therefore, I would like to accep this revised manuscript as it is.

Thanks.

Comments on the Quality of English Language

The authors have put their sincerely efforts into revising the manuscript. Therefore, I would like to accep this revised manuscript as it is.

Thanks.

Reviewer 3 Report

Comments and Suggestions for Authors

The authors have revised this manuscript according to the reviewer's comments.